# Simplifying the screening of gestational diabetes by maternal age plus fasting plasma glucose at first prenatal visit: A prospective cohort study

Yi-Yun Tai[1], Chien-Nan Lee[1], Chun-Heng Kuo[2], Ming-Wei Lin[1], Kuan-Yu Chen[3], Shin-Yu Lin[1]*, Hung-Yuan Li[4]*

1 Department of Obstetrics and Gynecology, National Taiwan University Hospital, Taipei, Taiwan,
2 Department of Internal Medicine, Fu Jen Catholic University Hospital, New Taipei City, Taiwan, 3 Taiwan Department of Internal Medicine, ANSN clinic, Hsinchu, Taiwan, 4 Department of Internal Medicine, National Taiwan University Hospital, Taipei, Taiwan

☯ These authors contributed equally to this work.
* lin.shinyu@ntu.edu.tw (SYL); larsli@ntuh.gov.tw (HYL)

**Data Availability Statement:** All relevant data are within the manuscript and its Supporting Information files.

## Abstract

### Aim

The addition of maternal age to fasting plasma glucose (FPG) at 24–28 gestational weeks improves the performance of GDM screening as maternal age increases. However, this method delays the diagnosis of GDM. Since FPG at the first prenatal visit (FPV) is a screening option for pre-existing diabetes, we evaluated the performance of age plus FPG at the FPV to reduce the need for the OGTT.

### Methods

Pregnant women were recruited consecutively in 2013–2018 (the training cohort) and 2019 (the validation cohort). We excluded women with twin pregnancies, unavailable FPG at the FPV or OGTT data, pre-pregnancy diabetes, or a history of GDM. All participants underwent FPG and haemoglobin A1c (HbA1c) at the FPV and received 75-g OGTT at 24–28 gestational weeks if FPG at the FPV was <92 mg/dL. GDM was diagnosed by the IADPSG criteria. Two algorithms were developed with the cutoffs determined when the percentage requiring OGTT (OGTT%) was the lowest and the sensitivity was ≥90%.

### Results

The incidence of GDM increased with age. The "FPG at the FPV" algorithm reduced OGTT% to 68.8% with the FPG cutoff at 79 mg/dl. The "age plus FPG at the FPV" algorithm, with the cutoff of 114, further reduced OGTT% to 58.3%, with the sensitivity of 90.7% (9.3% GDM missed) and the specificity of 100%. These findings were replicated in the validation cohort.

**Funding:** This work is supported in part by a grant (MOST 103-2314-B-002-157-MY2) from the Ministry of Science and Technology, Taiwan, and a grant (NTUH.105-S3192) from National Taiwan University Hospital, Taiwan.

**Competing interests:** The authors have declared that no competing interests exist.

## Conclusions

Screening GDM by maternal age plus FPG at the FPV can reduce OGTT%, especially in populations with a significant proportion of pregnant women with advanced ages.

## Introduction

For the diagnosis of gestational diabetes mellitus (GDM), the International Association of Diabetes and Pregnancy Study groups (IADPSG) and the World Health Organization (WHO) recommends the use of 75-g oral glucose tolerance test (OGTT) for all pregnant women at 24–28 weeks of gestation, which is also a diagnostic option suggested by the American Diabetes Association (ADA) in addition to the two-step method [1–3]. Since the OGTT is burdensome for the pregnant women and the healthcare system, some screening methods have been developed, such as risk factor-based models or the 1-hour 50g glucose challenge test used in the two-step method [4]. In addition, some studies have reported fasting plasma glucose (FPG)-based screening methods aimed at reducing the use of OGTTs [5–7]. However, these FPG-based screening methods have generally been targeted at 24–28 weeks of gestation instead of early pregnancy, and the use of an FPG-based method at 24–28 weeks essentially requires two steps to diagnose or exclude GDM [5–8].

On the other hand, the IADPSG recommends the measurement of FPG during early pregnancy to detect pre-existing diabetes and GDM [1, 9]; whereas the ADA suggests the screening of undiagnosed type 2 diabetes in the first prenatal visit (FPV) in women with risk factors by the standard criteria, such as FPG, hemoglobin A1c or even OGTT [3]. Therefore, pregnant women may have their FPG level tested at the FPV. Since a higher first-trimester FPG level is associated with an increased risk of GDM diagnosed at 24–28 weeks [10], the FPG at the FPV may be a good predictor to be used in the screening of GDM.

Although increasing maternal age is known to be a significant risk factor of diabetes mellitus (DM) [11], information regarding the relationship between age and GDM is reported relatively rarely. In most developed countries, women get pregnant at older ages [12–15]. Therefore, maternal age may be another important factor to be considered in the screening of GDM, especially in countries where women become pregnant at an older age.

Our previous study has shown that the rate of GDM with normal FPG increases as maternal age increases, and the addition of maternal age to FPG at 24–28 weeks of gestation can improve the performance of GDM screening [8]. However, if "age plus FPG at the FPV" can improve the performance of GDM screening remains unknown. Therefore, in this study, we aimed to evaluate the performance of "age plus FPG at the FPV" for the earlier screening of GDM as a means of reducing the need for the OGTT at 24–28 weeks.

## Research design and methods

### Data and sample collection

Two cohort studies were conducted for this project, including a prospective cohort study to develop the screening algorithms (the training cohort) and the other cohort study to validate the performance of the algorithms (the validation cohort). The training cohort study was conducted between January 2013 and June 2018 which recruited pregnant women who visited the obstetric clinic of National Taiwan University Hospital. The pregnant women in the validation cohort were recruited in 2019. The inclusion criterion for this study on screening for GDM

was singleton pregnancy delivering a phenotypically normal neonate at or after 28 weeks' gestation. We excluded pregnancies with twin pregnancies, or FPG at the first prenatal visit could not be obtained, or pre-pregnancy diabetes mellitus (either FPG $\geq$126 or HbA1c $\geq$ 6.5%), received OGTT in other clinic or hospital. Since it is reasonable for pregnant women with a history of GDM to undergo OGTT, we excluded women with a history of GDM from the analyses in this study (women with history of GDM, N = 35 in the training cohort, and N = 7 in the validation cohort). Eligible participants received blood tests for FPG and haemoglobin A1c (HbA1c) at the FPV after fasting for 8–10 hours. Women with FPG at the FPV less than 92 mg/dL received 75-g OGTT at 24–28 gestational weeks. The diagnosis of GDM was based on the recommendations of the IADPSG if FPG at the FPV $\geq$92 mg/dL, or if one of the results from the OGTT exceeded the cutoffs, including FPG $\geq$92 mg/dL, plasma glucose at 1h $\geq$180 mg/dL, or plasma glucose at 2h $\geq$153 mg/dL [1]. The clinicians were not blinded to the results of FPG at FPV and OGTT. Their clinical characteristics were acquired by questionnaires, such as age, parity, pre-pregnancy BMI, history of GDM and polycystic ovary syndrome (PCOS), family history of diabetes (defined as first degree relative with diabetes mellitus), etc. All participants signed informed consent before enrollment, and the institutional review board of National Taiwan University Hospital reviewed and approved the study protocol.

## Statistical analysis

Categorical variables were shown as numbers (percentages), and continuous variables with normal distribution were presented as means ± standard deviation (SD). Statistical significance between the GDM and non-GDM groups were analyzed by Student's t test and chi-squared test according to the nature of variables. Logistic regression models were used to estimate the odds ratios of GDM for various risk factors. Variables associated with GDM in univariate logistic regression analyses were included in the multiple logistic regression analysis, including age, FPG at FPV, HbA1c, family history of DM, and pre-pregnancy BMI. Then, two algorithms were constructed by the variables independently associated with GDM in the multivariate logistic regression model, including age and FPG at FPV. One algorithm used FPG at the first prenatal visits alone, and the other included both age and FPG at the first prenatal visits. To search for the optimal cutoffs to identify those who do not need further OGTT in the second trimester, we calculated the performance of each algorithm. The percentage of women requiring OGTT (OGTT%) was calculated as follows: (the number of women whose FPG at FPV<92 mg/dl and FPG at the FPV or age plus FPG at the FPV $\geq$ the cutoff value) / (the number of whole population). In both algorithms, pregnant women were diagnosed as GDM either by FPG at the FPV$\geq$92 mg/dL or the results of OGTT. Therefore, the false positive rate (FPR) was 0% for both algorithms. By definition, specificity equals to 1 –FPR, which means the specificity in all algorithms should ideally be 100%, no matter which cutoffs were chosen. Therefore, the determination of the optimal cutoffs was a trade-off between sensitivity and OGTT%. The optimal cutoff was chosen when the sensitivity was greater than 90% and the OGTT% was the lowest. We then simulated the relationship between the percentage of pregnant women older than 35 years and OGTT% according to these two algorithms. For every cutoff to exclude GDM, the prevalence of GDM, OGTT% and sensitivity values in women younger than 35 years (prevalence$_{<35y/o}$, OGTT%$_{<35y/o}$, and sensitivity$_{<35y/o}$) and in women older than 35 years (prevalence$_{\geq35y/o}$, OGTT%$_{\geq35y/o}$, and sensitivity$_{\geq35y/o}$) were calculated separately. The OGTT% and sensitivity in the whole population were calculated assuming the percentage of women older than 35 years (percentage$_{\geq35y/o}$) were 0%, 10%, 20%, etc., to 100%. For each percentage$_{\geq35y/o}$, OGTT% in the whole population was calculated as follows: (percentage$_{\geq35y/o}$ * OGTT%$_{\geq35y/o}$ + (1—percentage$_{\geq35y/o}$) * OGTT%$_{<35y/o}$). For each percentage$_{\geq35y/o}$,

sensitivity in the whole population was calculated as follows: (percentage$_{\geq 35y/o}$ * prevalence$_{\geq 35y/o}$ * sensitivity$_{\geq 35y/o}$ + (1—percentage$_{\geq 35y/o}$) * prevalence$_{<35y/o}$ * sensitivity$_{<35y/o}$) / (percentage$_{\geq 35y/o}$ * prevalence$_{\geq 35y/o}$ + (1—percentage$_{\geq 35y/o}$) * prevalence$_{<35y/o}$). For each percentage of women older than 35 years, the optimal cutoff was determined when the sensitivity was above 90% and the OGTT% was the lowest according to the algorithm. A two-tailed p-value below 0.05 was considered significant. Stata/SE 14.0 for Windows (StataCorp LP, College Station, TX) was used for statistical analyses.

## Result

### The training cohort and the validation cohort

A total of 1065 women who had their FPV between January 2013 and June 2018 were enrolled consecutively in the training cohort and another 151 women were included in the validation cohort from January 2019 to December 2019. Of the 1065 women who agreed to join our research, 55 women were excluded because of twin pregnancies, history of GDM, or pre-existing diabetes. We also excluded 419 women who were recruited at the second trimester and their FPG at the FPV could not be obtained. Another 79 women were excluded because they received OGTT in other clinic or hospital and the data were not available. As a result, 512 women were included in our study. A flowchart of the study population is shown in Fig 1. Besides, another 151 women were included from January 2019 to December 2019 as the validation cohort. All the study subjects in the training and validation cohorts were Chinese Han. In S1 Table, clinical characteristics between the 512 women with FPG and HbA1c at FPV and

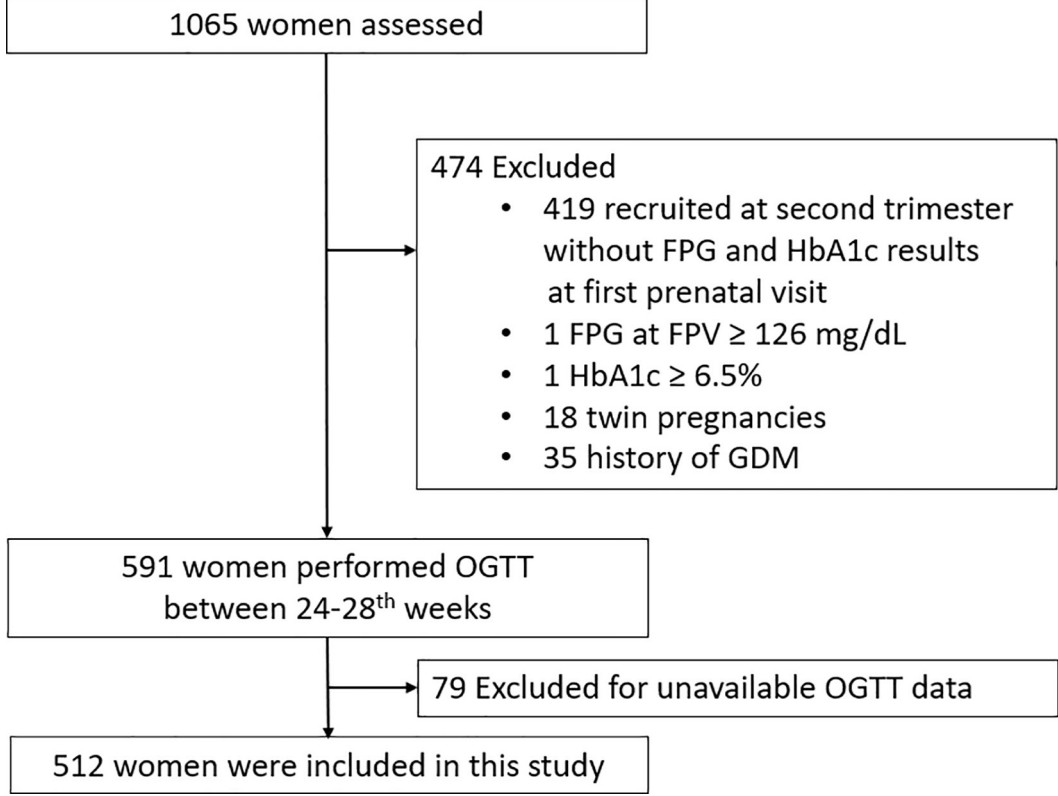

**Fig 1. The flow chart for the inclusion and exclusion of the study population.**

the 419 women without FPG and HbA1c at FPV were compared. There was no significant difference in clinical characteristics between these two groups.

Table 1 displays the clinical and obstetric characteristics of the women with GDM and the women without GDM in the training cohort and validation cohort. In the training cohort, the incidence of GDM was 14.6%. The mean gestational age of women at FPV was 10 weeks. In the training cohort, the women were older and the numbers of women ≥35 years were significantly higher in the GDM group than that in the non-GDM group. The women with GDM in the training cohort and in the validation cohort had higher BW, BMI, and higher levels of FPV and HbA1c. Besides, we have compared clinical characteristics in women with or without GDM between the training cohort and validation cohort. Women without GDM in the validation cohort had a slightly higher FPG and slightly lower HbA1c than women without GDM in the training cohort. In the women with GDM, 1-hour plasma glucose during OGTT at 24–28 gestational weeks were lower in the validation cohort than that in the training cohort.

## The incidence of GDM by age

As shown in Fig 2, the incidence of GDM increased with age. The incidence of GDM was 8.9% for the age group below 30 years, 15.4% for the age group 30–34 years, 20.2% for the age group 35–39 years, and 34.8% for the age group above 40 years. In addition, the level of FPG at the FPV also increased slightly with age (S1 Fig). The FPG at the FPV was also significantly higher in women with old age.

**Table 1. Clinical characteristics and laboratory test results in pregnant women with and without Gestational Diabetes Mellitus (GDM) in the training and validation cohort.**

| | The training cohort | | | The validation cohort | | |
|---|---|---|---|---|---|---|
| Baseline characteristics | Non-GDM | GDM | P value | Non-GDM | GDM | P value |
| | N = 437 (85.4%) | N = 75 (14.6%) | | N = 125 (80.2%) | N = 26 (17.2%) | |
| Age (years) | 33.5 (4.1) | 35.1 (4.1) | **0.002** | 33.3 (3.6) | 33.8 (2.9) | 0.55 |
| Age≥35 (N, %) | 190 (43.5%) | 42 (56%) | **0.04** | 46 (36.8%) | 11 (42.3%) | 0.32 |
| Nulliparous (N, %) | 142 (32.5%) | 26 (34.7%) | 0.68 | 46 (36.8%) | 13 (50%) | 0.48 |
| Gestational age at the FPV (weeks) | 10.1 (1.9) | 10.3 (2.2) | 0.81 | 10 (1.5) | 9.9 (1.5) | 0.58 |
| Family history of DM (N, %) | 96 (22%) | 26 (34.7%) | **0.02** | 21 (16.8%) | 5 (19.2%) | 0.81 |
| History of PCOS | 16 (3.7%) | 3 (4%) | 0.89 | 7 (5.6%) | 3 (11.5%) | 0.27 |
| History of macrosomia | 2 (0.5%) | 2 (2.7%) | 0.05 | 0 | 0 | - |
| Pre-pregnancy BW (kg) | 55.3 (8.4) | 57.8 (14) | **0.03** | 55.2 (8.1) | 60.2 (8.5) | **0.005** |
| Pre-pregnancy BMI (kg/m2) | 21.9 (3.3) | 22.9 (5.3) | **0.02** | 21.9 (3.2) | 23.7 (3.5) | **0.02** |
| GWG at 24–28 gestational weeks (kg) | 6.3 (3.7) | 6.3 (2.7) | 0.4 | 6.1 (3.2) | 6.2 (2.3) | 0.92 |
| Laboratory test results at the first prenatal visit | | | | | | |
| FPG (mg/dL) | 81.1 (5) | 88 (7.4) | **<0.001** | 82.8 (4.7)* | 90.9 (7.2) | **<0.001** |
| HbA1c (%) | 5.2 (0.2) | 5.4 (0.3) | **<0.001** | 5.1 (0.3)* | 5.3 (0.3) | **<0.001** |
| Glucose level during OGTT at 24–28 gestational weeks | | | | | | |
| FPG during OGTT (mg/dL) | 77.9 (5) | 82.3 (6.9) | **<0.001** | 78.8 (5) | 85.2 (5.4) | **<0.001** |
| 1hPG (mg/dL) | 128.2 (24.3) | 164.7 (29.5) | **<0.001** | 127.6 (27.3) | 149 (34.3)† | **<0.001** |
| 2hPG (mg/dL) | 109.9 (19.9) | 150.2 (29.4) | **<0.001** | 108.6 (21.3) | 135.6 (39.7) | **<0.001** |

Mean (standard deviations) or N (%) were shown.

BMI, body mass index; BW, body weight; DM, diabetes mellitus; FPG, fasting plasma glucose; FPV, first prenatal visit; GDM, gestational diabetes mellitus; GWG, gestational weight gain; HbA1c, hemoglobin A1c; OGTT, oral glucose tolerance tests; FPG during OGTT, fasting plasma glucose during oral glucose tolerance tests; PCOS, Polycystic ovary syndrome; 1hPG, 1-hour plasma glucose during oral glucose tolerance tests; 2hPG, 2-hour plasma glucose during oral glucose tolerance tests.

* p<0.05 vs. women without GDM in the training cohort.

† p<0.05 vs. women with GDM in the training cohort.

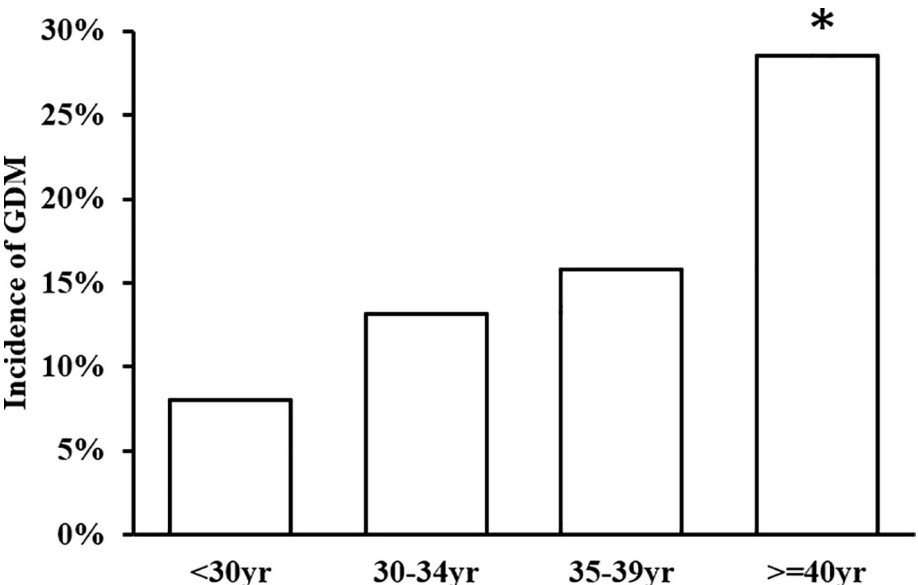

**Fig 2. The incidence of Gestational Diabetes Mellitus (GDM) by age group.** * p for trend across all ages <0.001.

## The development of the screening algorithms for GDM

Age and FPG at the FPV were the independent predictors of GDM in multivariate logistic model (S2 Table). Since the odds ratios and regression coefficients for age and FPG were similar, "age plus the FPG" was used to develop a screening algorithm for GDM. Two screening algorithms were constructed in women with FPG < 92 mg/dl, which used "FPG at the FPV" and "age plus FPG at the FPV" to exclude GDM. As for the optimal cutoffs, the performances of different cutoffs for "FPG at the FPV" and "age plus FPG at the FPV" in excluding GDM were calculated. In Table 2, as the cutoff increased, both the use of OGTT (OGTT%) and sensitivity decreased. Therefore, we chose cutoffs with a sensitivity ≥90% as the optimal cutoffs. The optimal cutoff value for FPG at the FPV in the "FPG at the FPV" algorithm was 79 mg/dl. Besides, the optimal cutoff value for age plus FPG at the FPV in the "age plus FPG at the FPV" algorithm was 114. By the cutoff of 114, if a 30-year-old woman whose FPG at the FPV was greater than 84 mg/dl, she should receive OGTT by the "age plus FPG at the FPV" algorithm, since her age plus FPG at FPV exceeded 114. Similarly, if a 35-year-old woman whose FPG at the FPV was greater than 79 mg/dl, then she should receive OGTT.

The flowchart of the algorithms to screen GDM were shown in Fig 3. In both algorithms, 29 pregnant women (5.7%) were diagnosed with GDM at FPV because their FPG was greater

**Table 2. Performance for different cutoffs to screen Gestational Diabetes Mellitus (GDM) by the "FPG at the FPV" algorithm and "age plus FPG at the FPV" algorithm.**

| FPG at the FPV | | | | Age plus FPG at the FPV | | | |
|---|---|---|---|---|---|---|---|
| Cutoffs to exclude GDM | OGTT (%) | Sensitivity (%) | Specificity (%) | Cutoffs to exclude GDM | OGTT (%) | Sensitivity (%) | Specificity (%) |
| 77 | 78.1% | 96% | 100% | 112 | 66.2% | 93.3% | 100% |
| 78 | 72.9% | 96% | 100% | 113 | 62.1% | 90.7% | 100% |
| **79** | **68.8%** | **92%** | **100%** | **114** | **58.3%** | **90.7%** | **100%** |
| 80 | 63.8% | 89.3% | 100% | 115 | 53.5% | 89.3% | 100% |

FPG, fasting plasma glucose; FPV, first prenatal visit; GDM, gestational diabetes mellitus; OGTT, oral glucose tolerance test.

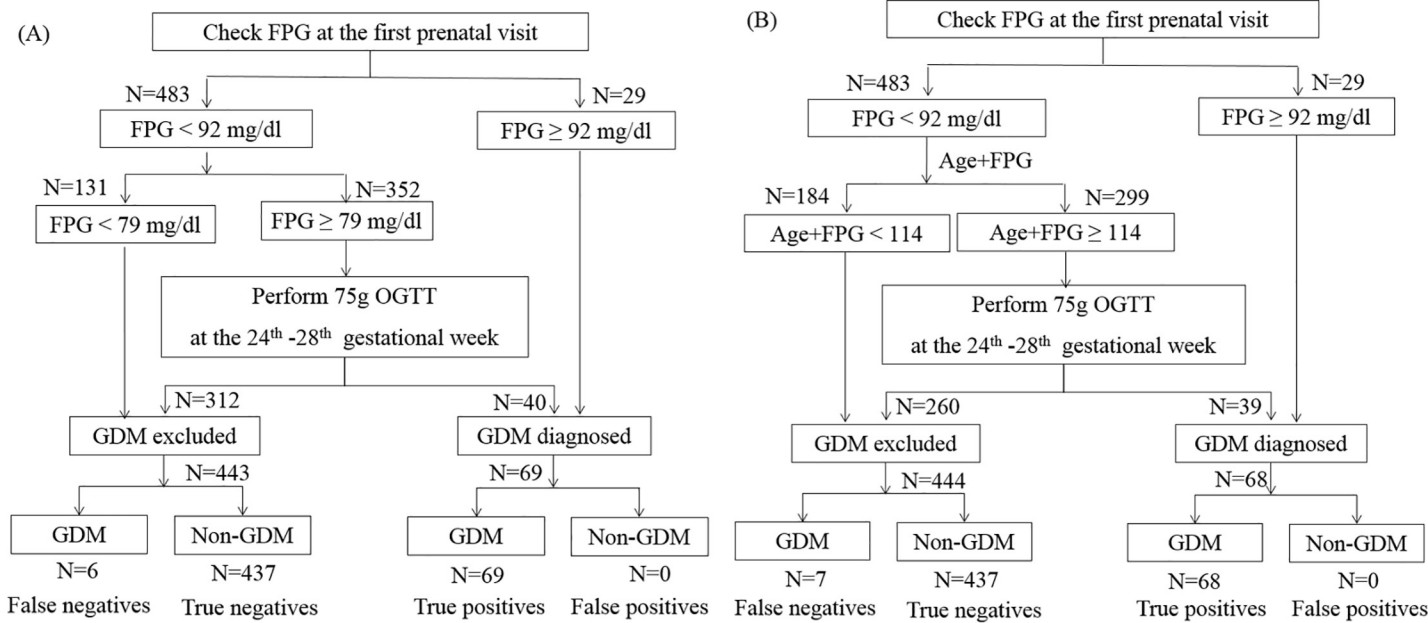

**Fig 3.** Algorithms to screen gestational diabetes mellitus (GDM) by (A) fasting plasma glucose (FPG) at the first prenatal visit (FPV) and (B) age plus FPG at the FPV. The percentages of pregnant women in different paths are shown. The unit for age is years and the unit for FPG is mg/dL. OGTT, oral glucose tolerance tests; GDM excluded, pregnant women who were diagnosed as not having GDM; GDM diagnosed, pregnant women who were diagnosed as GDM.

than or equal to 92 mg/dL. In algorithm of "FPG at the FPV" shown in Fig 3(A), if FPG at the FPV was below 79 mg/dL, GDM was excluded, and no OGTT was needed. If FPG was between 79 and 91 mg/dl, OGTT was recommended to confirm the diagnosis of GDM. The other "age plus FPG at the FPV" algorithm was shown in Fig 3(B). Among women with FPG <92 mg/dl, if "age plus FPG at the FPV" was below 114, GDM was excluded; if "age plus FPG at the FPV" was greater than or equals to 114, OGTT was suggested. In the training cohort, we found that the percentage of women who required OGTT was reduced to 68.8% by the "FPG at the FPV" algorithm. Compared to the "FPG at the FPV" algorithm, the "age plus FPG at the FPV" algorithm further reduced OGTT% to 58.5%. To better illustrate how the performance of the algorithms were calculated, we have added the numbers of women (N) in both algorithms (S2 Fig). The calculation of the sensitivity and the specificity for "the FPG at the FPV algorithm" with the cutoff at 79 mg/dl were illustrated in the S3 Table. The numbers were derived from S2(A) Fig. In the "FPG at the FPV" algorithms, GDM was diagnosed by FPG at the FPV ≥ 92 mg/dl (N = 29) or the results of OGTT when FPG at FPV ≥79 mg/dl (N = 40). FPG at the FPV ≥92 mg/dl and the results of OGTT are both diagnostic criteria of GDM by the IADPSG. No matter what the cutoff was chosen, the results of OGTT could correctly diagnose GDM. Therefore, there was no false positive by the algorithm, and the specificity of the algorithm was 100%. On the other hand, the "FPG at the FPV" algorithm excluded GDM if FPG <79 mg/dl (N = 131) or by the results of OGTT when FPG at FPV ≥79 mg/dl (N = 312). There were 6 women with GDM by the IADPSG criteria who were classified as "GDM excluded" (false negatives). Therefore, the sensitivity was 69/(6+69), which was equal to 92%. Since the results of OGTT could always correctly diagnose GDM by the IADPSG criteria, the false negatives came from women with FPG <79 mg/dl. If the FPG at the FPV cutoff was higher, there would be more false negatives (lower sensitivity) and less OGTT performed. The performance of the "age plus FPG at the FPV" algorithm was calculated similarly. Similarly, the OGTT% was decreased to 73.6% by

**Table 3. Performance of "FPG at the FPV" algorithm and "age plus FPG at the FPV" algorithm to screen Gestational Diabetes Mellitus (GDM) in the training cohort and in the validation cohort.**

| Algorithm | Threshold to exclude GDM | Training cohort | | | Validation cohort | | |
|---|---|---|---|---|---|---|---|
| | | OGTT (%) | Sensitivity (%) | Specificity (%) | OGTT (%) | Sensitivity (%) | Specificity (%) |
| FPG at the FPV | < 79 | 68.8 | 92 (83.4–97) | 100 | 72.8 | 89 (62.1–96.8) | 100 |
| "Age plus FPG at the FPV" | < 114 | 58.5 | 90.7 (81.7–96.2) | 100 | 59.6 | 92.3 (74.9–99.1) | 100 |

Estimates (95% confidence interval) were shown.

FPG, fasting plasma glucose; FPV, first prenatal visit; GDM, gestational diabetes mellitus; OGTT, oral glucose tolerance tests; FPG is in conventional units (mg/dL).

the "FPG at the FPV" algorithm and 61.5% by the "age plus FPG at the FPV" algorithm in the validation cohort (Table 3).

## The impact of age

To evaluate the impact of maternal age on OGTT%, we created several simulated populations with different percentages of pregnant women older than 35 years. As shown in Fig 4, when the percentage of women ≥35 years was between 30% and 60%, the screening algorithm with an "age plus FPG at the FPV" cutoff could reduce OGTT% while maintaining a sensitivity ≥90% and a specificity of 100%, compared with the screening algorithm using FPG at FPV alone.

## Discussion

In the present study, we found that the incidence of GDM increased with age. The "FPG at the FPV" algorithm could reduce OGTT% to 68.8% with the FPG cutoff at 79 mg/dl. The "age plus FPG at the FPV" algorithm, with the cutoff of 114, further reduced OGTT% to 58.3%, with the sensitivity of 90.7% (9.3% GDM missed) and the specificity of 100%. These findings were replicated in the validation cohort. Besides, we also shown that when the percentage of women ≥35 years was between 30% and 60%, the screening algorithm with an "age plus FPG at the FPV" cutoff could reduce OGTT%, compared with the screening algorithm using FPG at FPV alone.

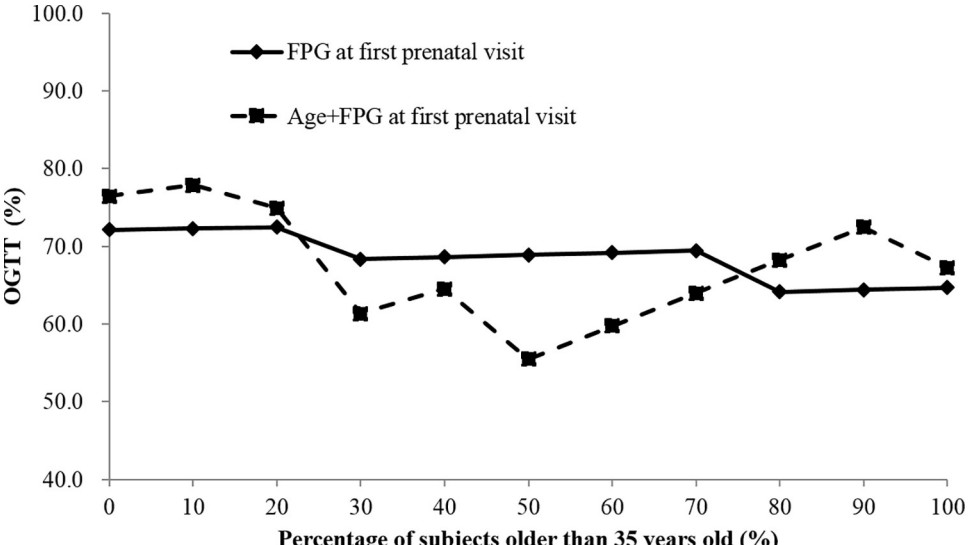

**Fig 4. The impact of maternal age on the percentage of OGTT needed (OGTT%) in simulated populations with different percentages of pregnant women older than 35 years.**

Age is an important determinant for insulin resistance and blood glucose levels. In non-pregnant adults, plasma glucose levels increase with age [16]. Several studies have shown that glucose tolerance declines progressively with age [17–24]. Aging-related glucose intolerance is more prominent in the third decade and continues throughout adulthood [25, 26]. However, the association has not been investigated in pregnant women in the literature. In the present study, we observed that FPG at the FPV increased by maternal age, and age was strongly associated with the prevalence of GDM. Our findings suggest that aging may have a similar effect on insulin resistance in pregnant women. Since women tend to get pregnant at older ages in most developed countries, advanced age in pregnancy may have greater impact on the risk of GDM in these countries.

OGTT is the "gold standard" to diagnose GDM. However, it is time-consuming for both patients and laboratories [27]. Therefore, many studies have proposed different screening methods to reduce the need of OGTT, and FPG is the most frequent studied screening tool since its measurement is easy, inexpensive, and reliable. Poomalar et al. have reported that FPG at 22–28 weeks is an effective screening tool for GDM [7]. With the cutoff value of 85 mg/dL, the sensitivity was 88% and the specificity was 95%. In our study, a lower FPG at the FPV cutoff value (79 mg/dL) had the best performance, with a sensitivity of 90% and a specificity of 100%. In our previous report, FPG at 24–28 gestational weeks has been shown to be a useful screening method, and the optimal cutoff value was 73 mg/dL [8]. Since FPG decreases gradually during the first half of pregnancy [28], the difference in the timing to test FPG may explain the different optimal cutoffs found in these two studies. Besides, using FPG at 24–28 weeks as the screening method may delay the diagnosis of GDM; whereas this can be avoided by using FPG at FPV. Furthermore, FPG at FPV can also be used to identify women with undiagnosed pre-pregnancy DM, as recommended by the IADPSG [1].

Our study suggests a potential way to predict GDM at the time of the first prenatal visit. We found that age and FPG at the FPV were the independent predictors for the development of GDM, which was supported by a report in the literature [29]. However, there is another study which evaluated the performance of FPG at FPV to predict GDM, and performance was not as good as our results [28]. There are some differences between that study and the present study. First, the diagnostic criteria of GDM were different. In their study, GDM was diagnosed when any one of the following value was met or exceeded during the 75g OGTT at 24–28 weeks, including fasting plasma glucose $\geq$92 mg/dL, 1-hour plasma glucose $\geq$180 mg/dL or 2-hour plasma glucose, $\geq$153 mg/dL. FPG at FPV $\geq$92 mg/dL was not a criterion to diagnose GDM. In our study, GDM was diagnosed according to the IADPSG criteria, which means that women with FPG at FPV $\geq$92 mg/dL, or data from OGTT exceeds the above-mentioned cutoffs were diagnosed GDM. As a result, the false positive rate was 0% in our algorithm, no matter which cutoffs were chosen. By definition, specificity equals to 1 –false positive rate. Therefore, the specificity in the algorithm is always 100%. In addition, although the mean age of the cohort in that paper was not described, we believe that it would not be high, since women often get pregnant at relatively young ages in China. In contrast, the mean age in our cohort was 33.7 years. All these differences may lead to the different results in these two studies. In addition, the NICE guidelines do not recommend the use of FPG to assess risk of developing GDM [30]. However, the NICE guidelines suggest a different criteria to diagnose GDM, ie. fasting plasma glucose greater than 100 mg/dl or 2-hour plasma glucose greater than 140 mg/dl. Therefore, the difference in the diagnostic criteria may contribute to the discrepancies between the recommendation from the NICE guidelines and the present study.

The strength of this study is that the algorithms used to screen GDM proposed in the study are simple, practical, and can be used clinically. The study was able to not only compare the predictive ability of age plus FPG at the FPV for GDM diagnosis but also to

describe the potential impact for implementation of the algorithm in clinical practice. Third, a simpler model with fewer variables would facilitate its implementation in daily practice, especially in under-resourced and overcrowded settings. "Age plus FPG at the FPV" showed good test characteristics and could thus serve as a screening option to decrease resource and time waste due to the excessive use of OGTTs. Furthermore, the mean gestational age of women at FPV in our study is 10 weeks. Identifying first-trimester biomarkers could serve both to diminish the need for provocative testing in all pregnant women and allow for early intervention to improve outcomes or prevent GDM. However, the study has several limitations. Firstly, all the study subjects were Han Chinese. Studies on other ethnic groups should be done to see if the findings could be generalized. Secondly, pregnant women with FPG at FPV $\geq$ 92 mg/dl and <126 mg/dl were diagnosed as GDM according to the IADPSG guidelines, in addition to the criteria by the results of OGTT at 24-28th gestational weeks. Since the diagnostic criteria for GDM varied in different countries [31], we have to emphasize that the conclusions of the present study are only applicable when GDM is diagnosed by the IADPSG criteria, which requires that all women have a fasting glucose test early in pregnancy. This may be difficult logistically and unpleasant for women. Besides, there were some small differences, such as FPG at the FPV, HbA1c, and 1-hour plasma glucose during OGTT, between women with or without GDM in the training cohort and in the validation cohort. Although the differences between these groups were small, those differences still may lead to the different performance. It should be noted when interpreting the result. Thirdly, the sample size is relatively small. Further studies with larger sample size are needed to verify the findings in this study.

In conclusion, FPG at the FPV and the prevalence of GDM increase by age. By using maternal age plus FPG at the FPV, we suggest a way to diagnose GDM with reduced number of people taking OGTT. This may be especially useful in populations with a significant proportion of women getting pregnant until older ages. Our results provide an alternative screening strategy to previously published algorithms using the combination of various risk factors of GDM, such as history of GDM, family history of diabetes, ethnicity, parity, and BMI [32, 33].

## Supporting information

**S1 Table. Clinical characteristics and laboratory test results at the first prenatal visit and OGTT result at 24–28 gestational weeks in pregnancy women with and without FPG and HbA1c at the first prenatal visit.**
(DOCX)

**S2 Table. The relationship between clinical characteristics at the first prenatal visit and of Gestational Diabetes Mellitus (GDM) at early pregnancy.**
(DOCX)

**S3 Table. The 2x2 table illustrating the calculation of the sensitivity and the specificity for "the FPG at the FPV algorithm" with the cutoff at 79 mg/dl.**
(DOCX)

**S1 Fig. Fasting Plasma Glucose (FPG) at the first prenatal visit by age group.**
(DOCX)

**S2 Fig.** Algorithms to screen gestational diabetes mellitus (GDM) by (A) fasting plasma glucose (FPG) at the first prenatal visit (FPV) and (B) age plus FPG at the FPV.
(DOCX)

## Acknowledgments

The authors would like to thank Mr. Chin-Mao Huang of National Taiwan University Hospital, and the staff of the eighth Core Lab, Department of Medical Research, National Taiwan University Hospital, Taipei, Taiwan, for technical and computing assistance.

## Author Contributions

**Conceptualization:** Yi-Yun Tai.

**Data curation:** Yi-Yun Tai.

**Formal analysis:** Yi-Yun Tai.

**Funding acquisition:** Chien-Nan Lee.

**Methodology:** Yi-Yun Tai, Hung-Yuan Li.

**Project administration:** Chun-Heng Kuo, Kuan-Yu Chen.

**Supervision:** Shin-Yu Lin.

**Validation:** Ming-Wei Lin.

**Writing – original draft:** Yi-Yun Tai.

**Writing – review & editing:** Yi-Yun Tai.

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
