## [Decision Letter · Decision Letter 0]

4 Feb 2020

PONE-D-20-00081

Simplifying the gestational diabetes screen by maternal age plus fasting plasma glucose at first prenatal visit: a prospective cohort study

PLOS ONE

Dear Dr. Lin,

Thank you for submitting your manuscript to PLOS ONE. After careful consideration, we feel that it has merit but does not fully meet PLOS ONE’s publication criteria as it currently stands. Therefore, we invite you to submit a revised version of the manuscript that addresses the points raised during the review process.

This manuscript is written in a very interesting and important subject area. 

There are three well-written reviews of the manuscript, each reviewer having a number of concerns (in a wide range of areas related to the manuscript). I believe that all the points raised by the reviewers are important, and would need to be dealt with robustly in any revision. Given the wide range of areas where the reviewers have highlighted concerns, I will not add any further necessary revisions, other than the point below about the need for language improvements.

We would appreciate receiving your revised manuscript by Mar 20 2020 11:59PM. To enhance the reproducibility of your results, we recommend that if applicable you deposit your laboratory protocols in protocols.io, where a protocol can be assigned its own identifier (DOI) such that it can be cited independently in the future. For instructions see: http://journals.plos.org/plosone/s/submission-guidelines#loc-laboratory-protocols

We look forward to receiving your revised manuscript.

Kind regards,

Clive J Petry, PhD

Academic Editor

PLOS ONE

Additional Editor Comments (if provided):

In addition to the various comments above and below, it is very important that in the revision the authors copy-edit the manuscript in order to improve the quality of English used. I would suggest using a professional copy-editing service, but if this is not possible, at least getting it checked out and corrected by a native English speaker.

Reviewers' comments:

Reviewer's Responses to Questions

**Comments to the Author**

1. Is the manuscript technically sound, and do the data support the conclusions?

Reviewer #1: Yes

Reviewer #2: No

Reviewer #3: Yes

2. Has the statistical analysis been performed appropriately and rigorously? 

Reviewer #1: Yes

Reviewer #2: No

Reviewer #3: No

3. Have the authors made all data underlying the findings in their manuscript fully available?

Reviewer #1: Yes

Reviewer #2: Yes

Reviewer #3: Yes

4. Is the manuscript presented in an intelligible fashion and written in standard English?

Reviewer #1: No

Reviewer #2: No

Reviewer #3: Yes

5. Review Comments to the Author

Reviewer #1: The authors present a prospective cohort study examining the relationship between age and fasting plasma glucose

for the diagnosis of gestational diabetes. The paper looks interesting, but needs major revisions in all chapters. In particular, the presentation must be changed and made clearer. I have the following comments and suggestions for revision:

Introduction

Page 5 Line 54

“The international diabetes study groups, including the International Association of

Diabetes and Pregnancy Study groups (IADPSG) and the American Diabetes Association

(ADA) have adopted the 75-g oral glucose tolerance test (OGTT) at 24–28 weeks of gestation

as a screening test for gestational diabetes mellitus (GDM).”

If you mention ADA and IADPSG, the correct reference must be reported both in the text and in the bibliography. In the references IADPSG has been reported with number 12.

Page 5 line 60

“However, these FPG-based screening methods have

generally been targeted at 24–28 weeks of gestation instead of early pregnancy, and the use of

an FPG-based method at 24-28 weeks essentially requires two steps to diagnose or exclude

DM.”

Please report the refereneces for this sentence.

Page 5 line 64

“On the other hand, FPG is nowadays routinely measured during early pregnancy to detect

pre-existing diabetes according to the IADPSG and ADA (4) diagnostic protocols.”

Please check the references and position them at the end of the sentence.

Page 5 line 69

“As such, the FPG level

at the FPV, which is easy to obtain, may be a good predictor to be used in screening GDM.”

Please report the references for this sentence.

Page 6 line 70

“Aging is thought to be one of the most important factors affecting the pathogenic

mechanisms associated with diabetes (6). However, despite the widely reported fact that

increasing age increases the prevalence of diabetes, information regarding the relationship

between age and GDM is reported relatively rarely. Furthermore, advanced maternal age is

also a growing trend. In most developed countries, women get pregnant at older ages (7-10).

Therefore, maternal age may be another important factor to be considered in GDM screening,

especially when women become pregnant at an older age.”

This paragraph does not seem to be written in correct English, please review the English grammar and the connection between the various sentences.

Research design and methods

Page 7

It is unclear how women with a blood glucose value ≥ 92 mg/dl are managed in the first trimester. OGTT ha salso been performed in these women?

According to the IADPSG guidelines, if the value is ≥ 92 and <126 mg/dl a diagnosis of gestational diabetes must be made and treated for this complication. What treatment were they subjected to? Did they perform OGTT even if in therapy?

If OGTT has been performed on women already on therapy this could be an important bias in your study.

Page 7 line 100

“Maternal blood samples were collected after 8–10 hours of overnight fasting

during and were used to determine HbA1c and FPG.”

In which trimester was this maternal blood sampesl performed? This sentence is unclear ans seems to be redundant. Please clarify.

Results

Page 8 – Page 14

Personally I find impossible to interpret the results as interesting as they seem. Text, tables and captions have been put together. Please rephrase the results completely and put them in an understandable way and following the submission guidelines.

Page 8 line 119

“A total of 547 patients were enrolled in the present study. A flowchart of the study

120 population is shown in S1 Figure.”

It is not clear from the description how the cohort was selected from the general population that affer to your Institute. I assume that more than 1065 women have been visited between January 2013 and June 2018. Was this a convenience sample? Were women recruited on certain days? This needs to be clarified as this a source of unrecognized bias that would need to be highlighted in the limitations.

Page 9 line 124

Table 1

Usually the tables must be presented in an attachment to the paper and not within the text. Please check the submission methods for this journal. Also personally I find the "All" column useless

Page 11 line 137

“As shown in Figure 1, the prevalence of GDM increased with age. 138

Fig 1. Prevalence of gestational diabetes mellitus (GDM) by age group. * p for trend

<0.001.”

This sentence at this point in the text has no meaning.

Page 9

In my opinion, some important factors are missing in interpreting the results: ethnicity, weight gain on the first visit and weight gain at the time of the OGTT.

If the weight gained during pregnancy is not available, please specify it.

Is it not possible that the women with increased fasting plasma glucose in 1st trimester had a superior weight gain compared to the other groups? Also, adding GWG to the multivariate model has clinical plausibility.

Also specify whether the women were of different or equal ethnic groups. This factor influenced the possibility of developing gestational diabetes.

Table S3

It is not personally clear whether a univariate or multivariate analysis was performed. These are very important data for your work and I would try to express them more clearly.

Discussion

Page 15 line 182

“Such studies have shown that plasma

glucose levels progressively increase with age.”

Please report references.

Page 17 line 221

“Our study demonstrates that GDM can be accurately predicted in early pregnancy based

on simple maternal clinical parameters available at the time of the first visit blood screening.”

The hypothesis of your study is that a first trimester screening for gestational diabetes based on risk factors could be performed.

This statement on a study carried out on 547 women, of whom only 98 diagnosed with gestational diabetes, can accurately predict gestational diabetes, seems very strong to me.

Pag 17 line 249

However, the limitations of this study is

that all the study subjects were Han Chinese, studies on other ethnic groups should be done to

see if the findings could be generalized.

Are there no weaknesses to the study? Please add.

Page 18 line

“In conclusion, FPG at the FPV and the prevalence of GDM increase by age. The

screening algorithm using maternal age plus FPG at the FPV can greatly simplify the

IADPSG diagnostic algorithm and reduce the use of the OGTT, especially in populations with

a significant proportion of women who become pregnant at older ages. The optimal cutoff

value for age plus FPG at the FPV is 115.”

Sea above, this conclusion is too strong for such a reduced study cohort. Furthermore in literature other methods are described to implement diabetes screening already in the first trimester as algorithms, maternal characteristics, ultrasound evaluation of maternal adipose tissue, etc… personally I would quote some.

Reviewer #2: The authors present a cohort study aimed at devising a strategy to decrease the need for OGTT at 24-28 weeks gestation based on the combination of maternal age and fasting plasma glucose (FPG) at the first prenatal visit (FPV). This is a subject that has practical clinical applicability and could be useful in clinical settings if proven safe and effective. Many studies have attempted to create predictive models in early pregnancy to predict mid trimester GDM. Most are complex and/or have poor sensitivity, specificity or both. My comments regarding the current study can be found below. The paper will benefit from additional grammatical revision

1. I don’t understand the subtitle: “Early screen of gestational diabetes: role of maternal age” appears to be redundant or inserted by mistake.

Abstract:

2. As below – FPG at FPV is not universally performed or recommended

3. 547 were not consecutively recruited as > 1000 were eligible and approached

Introduction

4. line 54-57: Inaccurate - ADA has allowed the OPTION of one step 75 gram OGTT but also allow the wo step 50 g GCT/100 gram GTT

5. FPG at first visit is NOT universally recommended by ADA – only in those with risk factors. Even in those the recommendation is to test using “standard criteria”

6. Line 64 – 66 again, FPG not universally performed at FPV or first trimester (likely the same)

7. Line 67-69 – sentence contradicts itself if “poorly predictive” of GDM in early pregnancy why would FPG at the FPV (which is also early pregnancy) be a “good predictor” ?

8. Line 70: Really? Age is more independently important than obesity, family history previous GDM etc?

Methods

9. If I understand correctly, women were recruited at FPV and were eligible if they were singleton, viable and had no known preexisting DM. Then they had FPG and A1C. If they had PDM after these tests they were excluded. All that remained had a OGTT at 24-28 weeks. This should be clarified and shown in a flow diagram with numbers as Fig 1 (not as supplementary material)

10. From figure S1 – I note that nearly half of the base population did not have FPG and A1C. In line 92 you say that “ALL pregnant women were done with FPG and hemoglobin A1c (HbA1c)…” Please clarify this. Was in fact FPG and A1C a standardized protocol at the time of the study? If so why did > 400 women not have this done. Were they perhaps different clinically than those that did? Were all 1065 approached and consented if singleton and no PDM? If so – did the 419 without FPG and A1C not consent? This is all confusing and needs to be clarified.

11. Was there a gestational age criteria for the FPV? If someone had a FPV at 20 weeks was she included? Please clarify

12. When defining BMI was prepregnancy BMI used? Early pregnancy?

13. Were the FPG and A1C results available to the clinicians? What was done if at FPV the fPG was > 5.1 or A1C as 5.9-6.4?

Results

14. Was gestational weight gain collected and if not – why?

15. Why is FPG of >91 at FPV (10 weeks mean) diagnostic of GDM (Fig 2)?? This is not standard. The IADPSG values are extrapolated from HAPO which used GTT values at 24-28 weeks. There are no outcome based validated thresholds for early GTT. Were these women treated (see comment 11)

16. Line 142-143: In addition to age, previous GDM was also independently associated with GDM OR 8.14. This is not mentioned. Also what about the effect of BMI? It was significant on the univariate analysis and it is mentioned in the legend (and in line 232 of the discussion) but not in the body of the table. What happened to this variable?

Discussion

17. Discussion overall VERY repetitive.

18. Line 227 – provide references for evidence that early identification of “GDM” leads to better outcomes

19. These results are not consistent with the study by Zhu et al DIABETES CARE, VOLUME 36, MARCH 2013. In this study even if FPG at FPV as < 4.1 8.1% had a subsequent diagnosis of GDM at 24-28 weeks. In addition they were able to show that overall – there was poor correlation between the FPG at FPV and the FPG at the OGTT. 100% specificity was achieved only > 5.7 mmol (100.8) and with corresponding low sensitivity. Please explain the difference between this (large) study’s results and yours. In addition discuss why FPG performs favorably as a screening tool in this study in contrast with what is known regarding the poor performance of early FPG as a screening tool for future GDM (See summary of evidence in NICE guidelines).

20. Missing references for other predictive models including recent: Zheng, T., Ye, W., Wang, X. et al. A simple model to predict risk of gestational diabetes mellitus from 8 to 20 weeks of gestation in Chinese women. BMC Pregnancy Childbirth 19, 252 (2019). https://doi.org/10.1186/s12884-019-2374-8.

21. Missing many of the other limitations:

a. The “missing” 419 women that did not have FPG and A1C at FPV. If you can show that their characteristics are not different from those that had testing then that will provide some support for the validity of the results

b. The lack of clinician blinding

c. Unlike the authors previous study – there is no validation cohort

d. No reporting of gestational weight gain

e. No reporting of perinatal outcomes. What are the outcomes for those missed by this strategy (90% sensitivity compared with 100% sensitivity with universal OGTT)

22. The conclusion is way too conclusive. The most you can say is that these data suggest an alternative method for early screening for GDM but needs to be prospectively validated in multiple ethnic cohorts and with appropriate reporting of perinatal outcomes. The authors should consider publishing only after external validation of their findings

Reviewer #3: The paper handles the intriguing matter of how to rationalise screening for GDM- increasing incidence of GDM due to demographic changes and diagnostic thresholds have led to increased burden on health care systems. This paper investigates a method of using risk-factor based screening, tailored by early pregnancy fasting glucose, to increase the diagnostic performance of the mid trimester OGTT, without too great an impact on missed GDM diagnoses.

Abstract

The abstract needs to state whether all women underwent fasting glucose screening (universal or risk factor based). The abstract also needs to describe what the OGTT eligibility criteria were (not including maternal age and fasting glucose)- this could be a brief statement just mentioning risk factor based or universal. It appears that 62% of pregnant women were OGTT screened at baseline: was this because certain thresholds of BMI or family history were used to determine eligibility for OGTT? Or did a large group of women decline screening by OGTT? The abstract needs to state which threshold was used for fasting glucose, and how this was determined, and which age threshold was applied- generally adding a little more numerical data on the paper’s findings. Also, please add the number of diagnoses of GDM missed when the number of OGTTs is reduced by the use of maternal age, and fasting glucose.

Introduction

Needs to specify what the current recommendations are of guidelines (e.g. ADA, WHO) regarding universal or risk-factor based screening or prognostic model guided screening. To my knowledge, the guidelines to not specify a preference-this would be good to point out in the introduction.

Using prognostic models is not a new approach.

Methods

Could benefit from specifying which variables were eligible for the prognostic model; did BMI or family history or obstetric history feature in the multivariable logistic model?

Also, needs to state that pregnant women who qualified for GDM diagnosis after just a fasting glucose, were not eligible to continue in the study.

Was PCOS diagnosis available in this study, or was history of macrosomia available? These are additional risk factors that are sometimes used to specify eligibility for OGTT.

The comment ‘the specificity for all models was 100%, regardless of the threshold used’ is likely due to the way the model was established. Specificity is the probability that, given that someone has no GDM, the test would indicate that she had no GDM: given that the authors were looking at the effect of reducing the number of OGTTs performed, they are not going to find false positives (so lower specificity). This ‘self-fulfilling prophecy’ could be more clearly explained in the last paragraph of the methods’.

It is unclear why the authors have focussed on only age and fasting glucose in their attempt to reduce the number of OGTTs, when more data is available. This needs to be elaborated.

Results

Given that almost half of women were excluded, it would be good to present a table with baseline characteristics of those with and without OGTT data available (so not twins or those with suspected pre-conception DM2), to establish generalisability of this study’s findings.

Figure 1 states it reports the prevalence, when I think the authors mean incidence. Please amend.

S2 Figure states that the ‘p for trend’ is starred, but does not specify for which comparison the star is presented: is this the trend across all ages, or the difference between highest and lowest age category.

Figure 2 presents the number of OGTTs in populations of varying age distributions. The models used to create this figure are not described in the methods, and must be added. Presumably, there would be some uncertainty (SD or 95% confidence interval) which could be better demonstrated in the figure.

In line 158, at the end of the results, the actual age threshold used need to be added to the text. Did the authors toggle the age threshold, as they did for fasting glucose?

The last sentence of the results (line 159-160) now reads: the addition of FPG could reduce OGTT from 62 to 52%, this is strictly speaking not correct. The reduction from the current policy describe in the methods, is 100% OGTT to 52% OGTT screening when maternal age>=35 years and FPG> =80 are applied as eligibility criteria.

The methods need to show whether women with a history of GDM were also excluded from OGTT screening if they were neither past 35 or had fasting glucose <80. I am assuming this is the case, but due to the rather large impact on chance of GDM, this needs to be explicitly stated.

Minor comments

Generally, the English is fine, although it could be improved by English language editing. Examples include (Introduction) ‘…higher first-trimester FPG levels in early pregnancy is actually poorly….’ Which should be ‘…are actually poorly…’;

and (line 92 methods) ‘….All pregnant women were done with FPG…’ Which should be ‘…all pregnant women underwent FPG …’

and (line 102 methods) ‘… Maternal blood samples without fasting were excluded…’ which should be ‘… we excluded maternal blood samples of participants who had not fasted before the blood sample was drawn…’

Supporting files flow chart ‘preformed’ which should be ‘performed’

6. PLOS authors have the option to publish the peer review history of their article (what does this mean?). If published, this will include your full peer review and any attached files.

Reviewer #1: No

Reviewer #2: No

Reviewer #3: Yes: Rebecca C. Painter

---

## [Author Response · Author response to Decision Letter 0]

19 Apr 2020

Please referred to the file "Response to Reviewers".

---

## [Decision Letter · Decision Letter 1]

6 May 2020

PONE-D-20-00081R1

Simplifying the screening of gestational diabetes by maternal age plus fasting plasma glucose at first prenatal visit: a prospective cohort study

PLOS ONE

Dear Dr. Lin,

Thank you for submitting your manuscript to PLOS ONE. After careful consideration, we feel that it has merit but does not fully meet PLOS ONE’s publication criteria as it currently stands. Therefore, we invite you to submit a revised version of the manuscript that addresses the points raised during the review process.

I agree with the reviewers that the various revisions made have improved the manuscript a lot. However there is a disagreement between the reviewers, both of whom also reviewed the original manuscript, over the merits of the revised version. Reviewer #1 is happy with the manuscript, which is a valid point. I feel that the comments of reviewer #2 also have merit, and therefore that a further round of revision needs to be performed. It would be wrong to accept the manuscript for publication when these issues have been raised. I think that points 2 and 5 of reviewer #2's comments need to be dealt with particularly robustly in a second revision, the other points should be easier to address. I have no additional comments and look forward to reading the second revision of the manuscript.

We would appreciate receiving your revised manuscript by Jun 20 2020 11:59PM. To enhance the reproducibility of your results, we recommend that if applicable you deposit your laboratory protocols in protocols.io, where a protocol can be assigned its own identifier (DOI) such that it can be cited independently in the future. For instructions see: http://journals.plos.org/plosone/s/submission-guidelines#loc-laboratory-protocols

We look forward to receiving your revised manuscript.

Kind regards,

Clive J Petry, PhD

Academic Editor

PLOS ONE

Reviewers' comments:

Reviewer's Responses to Questions

**Comments to the Author**

1. If the authors have adequately addressed your comments raised in a previous round of review and you feel that this manuscript is now acceptable for publication, you may indicate that here to bypass the “Comments to the Author” section, enter your conflict of interest statement in the “Confidential to Editor” section, and submit your "Accept" recommendation.

Reviewer #1: All comments have been addressed

Reviewer #2: (No Response)

2. Is the manuscript technically sound, and do the data support the conclusions?

Reviewer #1: Yes

Reviewer #2: No

3. Has the statistical analysis been performed appropriately and rigorously? 

Reviewer #1: Yes

Reviewer #2: No

4. Have the authors made all data underlying the findings in their manuscript fully available?

Reviewer #1: Yes

Reviewer #2: Yes

5. Is the manuscript presented in an intelligible fashion and written in standard English?

Reviewer #1: Yes

Reviewer #2: Yes

6. Review Comments to the Author

Reviewer #1: Thank you for having clearly answered all the questions and for the changes made. No further corrections are required.

Reviewer #2: I would like to thank the authors for the thoughtful revisions. I still have significant reservations regarding the data presented with regards to specificity and sensitivity of the models (see comment 5) General: Some additional grammatic editorial work is needed, especially in the parts that were revised.

1. Introduction line 75 : should be …FPG level is associated

2. From the new manuscript I now see that only those that had FPG at FPV of < 92 had a 24-28 week 75 gram OGTT. I thus assume that those with FPG > 91 were treated as GDM. Please clarify. If these women were in fact treated as GDM, this makes it very difficult to calculate the performance of age+FPG at FPV in diagnosing later GDM. In order to do this, FPG at FPV would be drawn in a blinded fashion (unless overt diabetes was found) and all would have an unblinded 75 gram OGTT at 24-28 weeks. Thus any conclusions from this study are only applicable to women with FPG < 92 at booking. This needs to be stated in the limitations as many countries do not view a FPG of >91 at FPV as synonymous with a diagnosis of GDM.

3. Methods Line 108: What does it mean the clinicians were “blinded to the results of the statistical analysis” The analysis was done retrospectively after the cohorts were formed so this would not be relevant. What is relevant, is to indicate that clinicians were not blinded to the FPG at FPV

4. Methods line 125 “One algorithm made use FPG…” revise the sentence

5. Methods lines 131 – 134, Discussion 304-306, Table 2 and Figure 3. I need to disagree with the statement that specificity was 100%. If one is doing a FPG and FPV (diagnosing GDM at FPV on all above arbitrary threshold) and then a GTT (diagnostic test) in all the remaining, then one is NOT doing screening but rather universal diagnostic testing with, by design, 100% sensitivity and 100% specificity in the entire cohort. Thus, do to the study design, the specificity was 100% in the cohort but this would not be true for the models using different thresholds for FPG at FPV. In your models I believe you are actually attempting to answer the question: how does FPV FPG below the diagnostic threshold perform as a screening test for 24-28 week GDM compared with FPG+Age at FPV? Based on Figure 3 the specificity is definitely not 100%. In both algorithms 59.9% and 49.4% that go on to have the OGTT based on the FPG or FPG+age cutoffs respectively, are found NOT to have GDM and thus are FALSE POSITIVES for the FPG screening test and the outcome of GDM. From the figure – there is no information on the false negatives i.e those below the threshold (FPG or FPG+Age) that were diagnosed with GDM. We know that number is not 0 as in the results (Table 2) the sensitivity is < 100%. One can design the algorithms to perform at a prespecified sensitivity or specificity (even a theoretical 100%) but always (unless you have found the perfect screening test when compared to the gold standard), high specificity will be at the expense of sensitivity and vice versa. Neither the false negatives or the false positives are correctly indicated in the figure or in the text and thus the interpretation of this study appears problematic. I think that in the algorithms the authors need to reflect on their assumptions regarding specificity and sensitivity or at least recheck the numbers. Once this is sorted out – reporting a ROC curve and the PPV and NPV would be useful.

6. Methods – line 140-143: language is unclear

7. Results Line 151-155 – you are repeating methodology – should not be in the results

8. Table 1: Ideally there would be a comparison of the characteristics of the two cohorts to demonstrate that over time there was no change in population characteristics

9. Results will be easier to follow if arranged with subheadings

10. Table 4. The outcome analysis. The comparison should not be between outcomes in women with and without GDM (as obviously the rate will be higher in GDM) but rather between rate of outcomes in each cohort compared with gold standard (IADPSG). Due to the way this study was designed the results are not representative of true perinatal risk as NO GDMs were undiagnosed and untreated, which is what would happen if using a first trimester screening test with < 100% sensitivity. Based on this this table can likely be omitted and replaced with a statement that “ The impact of adopting either algorithm needs to be evaluated in future prospective blinded trials”

11. Discussion: Line 328-330. Actually I do not believe that not having GWG at the FPV is significant. There is minimal weight gain in the first 10 weeks of pregnancy thus the impact would be minimal. Adjusting for GWG up until the 24-28 week GTT is relevant

7. PLOS authors have the option to publish the peer review history of their article (what does this mean?). If published, this will include your full peer review and any attached files.

Reviewer #1: No

Reviewer #2: No

---

## [Author Response · Author response to Decision Letter 1]

15 Jun 2020

Please refer to the attached files.

---

## [Decision Letter · Decision Letter 2]

23 Jun 2020

PONE-D-20-00081R2

Simplifying the screening of gestational diabetes by maternal age plus fasting plasma glucose at first prenatal visit: a prospective cohort study

PLOS ONE

Dear Dr. Lin,

Thank you for submitting your manuscript to PLOS ONE. After careful consideration, we feel that it has merit but does not fully meet PLOS ONE’s publication criteria as it currently stands. Therefore, we invite you to submit a revised version of the manuscript that addresses the points raised during the review process.

The manuscript has improved by the various revisions made. Sorry to have to ask for further revision but the reviewer still has a few very minor points that need addressing (mainly language-related, but an important one regarding the specificity, which is the most important point to deal with). 

Having been highlighted by the reviewer I think that there should be one final round of revision. I look forward to seeing a final version that, in particular, addresses the point about specificity.

We look forward to receiving your revised manuscript.

Kind regards,

Clive J Petry, PhD

Academic Editor

PLOS ONE

Reviewers' comments:

Reviewer's Responses to Questions

**Comments to the Author**

1. If the authors have adequately addressed your comments raised in a previous round of review and you feel that this manuscript is now acceptable for publication, you may indicate that here to bypass the “Comments to the Author” section, enter your conflict of interest statement in the “Confidential to Editor” section, and submit your "Accept" recommendation.

Reviewer #2: (No Response)

2. Is the manuscript technically sound, and do the data support the conclusions?

Reviewer #2: Partly

3. Has the statistical analysis been performed appropriately and rigorously? 

Reviewer #2: No

4. Have the authors made all data underlying the findings in their manuscript fully available?

Reviewer #2: Yes

5. Is the manuscript presented in an intelligible fashion and written in standard English?

Reviewer #2: No

6. Review Comments to the Author

Reviewer #2: The authors have provided responses to the previous comments and improved the paper. I have a few remaining issues:

After all the revisions, the paper needs additional English language editing (e.g “The pregnant women in the validation cohort was recruited..” “Their clinical characteristics were acquired by questionnaire and recorded such as age, parity...”)

Page 10 line 163 – repeats page 10 line 156-157. Repeat description of validation cohort

Page 16 line 238: TYPO “FPG at the FPG ≥92 mg/dl and the results of OGTT are both…”

Table 1: Title Suggest instead of ” Clinical characteristics and laboratory test results at the first prenatal visit and OGTT result at 24-28 gestational weeks in pregnancy women with and without gestational diabetes mellitus (GDM) in the training cohort and validation cohort” - should be simplified to “Clinical characteristics and laboratory test results in pregnant women with and without gestational diabetes mellitus (GDM) in the training and validation cohort” Also add n=XX under the column headings

From table 1 – you appropriately highlight the small differences in blood sugar results between the validation and derivation cohort. This should be mentioned as a possible limitation to the validation portion of the analysis.

The prevalence of GDM is quite high – overall 14.6%. Does this seem reasonable for a universally tested Taiwanese population? If not – please address.

The authors state in the discussion that “the algorithms used to screen GDM proposed in the study are simple, practical, and can be used clinically”. This is true but it needs to be mentioned that it requires that all women have a fasting glucose test early in pregnancy. This is often difficult logistically and unpleasant for women that are likely experiencing nausea and vomiting. This can be mentioned as a drawback to this strategy.

The discussion would usually start with a summary statement of the major findings. This is not the case here.

Finally, I need to come back to the statement that there are no false positives and thus the specificity is 100%. See 2X2 table for the screening strategy of FPG at FPV using a threshold of 79 in all women with FPG <92 at FPV and using the values from S2 Fig:

SEE UPLOADED DOCUMENT

7. PLOS authors have the option to publish the peer review history of their article (what does this mean?). If published, this will include your full peer review and any attached files.

Reviewer #2: No

---

## [Author Response · Author response to Decision Letter 2]

20 Jul 2020

We have revised according to reviewers’ suggestions, point by point. All the questions raised by the reviewers are carefully addressed in the response to reviewers.

---

## [Editor Report · Decision Letter 3]

23 Jul 2020

Simplifying the screening of gestational diabetes by maternal age plus fasting plasma glucose at first prenatal visit: a prospective cohort study

PONE-D-20-00081R3

Dear Dr. Lin,

We’re pleased to inform you that your manuscript has been judged scientifically suitable for publication and will be formally accepted for publication once it meets all outstanding technical requirements.

Kind regards,

Clive J Petry, PhD

Academic Editor

PLOS ONE
---

## [Editor Report · Acceptance letter]

6 Aug 2020

PONE-D-20-00081R3 

Simplifying the screening of gestational diabetes by maternal age plus fasting plasma glucose at first prenatal visit: a prospective cohort study 

Dear Dr. Lin:

I'm pleased to inform you that your manuscript has been deemed suitable for publication in PLOS ONE. Congratulations! Your manuscript is now with our production department. 

Kind regards, 

on behalf of

Dr. Clive J Petry 

Academic Editor

PLOS ONE